# Revisiting the stability of stochastic gradient descent: a tightness analysis

## Abstract

The technique of algorithmic stability has been used to capture the generalization power of several learning models, especially those trained with stochastic gradient descent (SGD). This paper investigates the tightness of the algorithmic stability bounds for SGD given by Hardt et al. (2016). We show that the analysis of Hardt et al. (2016) is tight for convex objective functions, but loose for non-convex objective functions. In the non-convex case we provide a tighter upper bound on the stability (and hence generalization error), and provide evidence that it is asymptotically tight up to a constant factor.

However, deep neural networks trained with SGD exhibit much better stability and generalization in practice than what is suggested by these (tight) bounds, namely, linear or exponential degradation with time for SGD with constant step size. We aim towards characterizing deep learning loss functions with good generalization guarantees, despite training using SGD with constant step size.

In this vein, we propose the notion of a Hessian Contractive (HC) region, which quantifies the contractivity of regions containing local minima in the neural network loss landscape. We provide empirical evidence that several loss functions exhibit HC characteristics, and provide theoretical evidence that the known tight SGD stability bounds for convex and non-convex loss functions can be circumvented by HC loss functions, thus partially explaining the generalization of deep neural networks.

## 1 Introduction

*Stochastic gradient descent* (SGD) has gained great popularity in solving machine learning optimization problems (Kingma & Ba, 2014; Johnson & Zhang, 2013). SGD leverages the finite-sum structure of the objective function, avoids the expensive computation of exact gradients, and thus provides a feasible and efficient optimization solution in large-scale settings (Bottou, 2012). The convergence and the optimality of SGD have been thoroughly studied (Ge et al., 2015; Rakhlin et al., 2012; Reddi et al., 2018; Zhou & Gu, 2019; Carmon et al., 2019a;b; Shamir & Zhang, 2013) .

In recent years, new research questions have been raised regarding SGD's impact on a model's generalization power. The seminal work (Hardt et al., 2016) tackled the problem using the *algorithmic stability* of SGD, i.e., the progressive sensitivity of the trained model w.r.t. the replacement of a single (test) datum in the training set. The stability-based analysis of the generalization gap allows one to bypass classical model capacity theorems (Vapnik, 1998; Koltchinskii & Panchenko, 2000) or weight-based complexity theorems (Neyshabur et al., 2017; Bartlett et al., 2017; Arora et al., 2018). This framework also provides theoretical insights into many phenomena observed in practice, e.g., the "train faster, generalize better" phenomenon, the power of regularization techniques such as weight decay (Krogh & Hertz, 1992), Dropout (Srivastava et al., 2014), and gradient clipping. Other works have applied the stability analysis to more sophisticated settings such as Stochastic Gradient Langevin Dynamics and momentum SGD (Mou et al., 2018; Chaudhari et al., 2019; Chen et al., 2018).

Despite the promises of this stability-based analysis, it remains open whether this framework can explain the strong generalization performance of deep neural networks in practice. Existing theoretical upper bounds of the stability (and thus, generalization) (Hardt et al., 2016) are ideal for strongly convex loss functions: the upper bound remains constant even as the number of training iterations increases. However, the same bound deteriorates significantly when we relax to more general and realistic settings. In particular, for convex (but not strongly convex) and non-convex loss functions, if

SGD has constant step size, then the upper bound grows linearly and exponentially with the number of training iterations. This bound fails to match the superior generalization performance of deep neural networks, and leads to the following question:

**Question 1:** *Can we find a better stability upper bound for convex or non-convex loss functions?*

In this paper, we first address the question above and investigate the tightness of the algorithmic stability analysis for stochastic gradient methods (SGM) proposed by (Hardt et al., 2016).

**R1**. We show in Theorem 1 that the analysis in (Hardt et al., 2016) is tight for convex and smooth objective functions; in other words, there is a convex loss function whose stability grows linearly with the number of training iterations, with constant step size ($\alpha_t = \alpha$) in SGD.

**R2**. We show that in Theorem 2 that for linear models, the analysis in the convex case can be tightened to show that $\epsilon_{\text{stab}}$ does not increase with $t$.

**R3**. In Theorem 3 we show that the analysis in (Hardt et al., 2016) for decreasing step size ($\alpha_t = O(1/t)$) is loose for non-convex objective functions by providing a tighter upper bound on the stability (and hence generalization error).

**R4**. The bound on the stability of SGD by (Hardt et al., 2016) is achieved by bounding the divergence at time $t$, defined as $\delta_t := \mathbb{E}||w_t - w_t'||$, where $w_t$ is the model trained on data set $S$ and $w_t'$ is the model trained on a data set $S'$ that differs from $S$ in exactly one sample. In Theorem 4 we provide evidence that our new upper bound in the non-convex case is tight, by showing a non-convex loss function whose divergence matches the upper bound for our divergence.

**R5**. Although it is not derived formally, the techniques in (Hardt et al., 2016) can be employed to show an exponential upper bound for non-convex loss functions minimized using SGD with constant-size step. In Theorem 5, we give evidence that this abysmal upper bound is likely tight for non-convex loss functions, by exhibiting a non-convex loss function for which the divergence $\delta_t$ increases exponentially.

Thus the only functions whose stability provably does not increase with the number of iterations when a constant step-size during SGD is employed, are strongly convex functions. However, a) it has been empirically observed that for deep neural network loss, near the local minima, the Hessians are usually low rank (Chaudhari et al., 2017; Yao et al., 2019), and b) neural networks trained with constant step-size SGD do generalize well in practice (Lin & Jegelka, 2018; Huang et al., 2017; Smith et al., 2017). Combined with our lower bounds on convex and non-convex functions, we seem to hit an obstacle on the way to explaining generalization using the stability framework.

**Question 2:** *What is it that makes constant-step SGD on deep learning loss function generalize well?*

Realizing the limitation of the current state of stability analysis, we investigate whether a stronger-than-convex, but weaker-than-strongly-convex assumption of the loss function can be made, at least near local minima. If we can show algorithmic stability near local minima, we can still show the stability using similar argument as (Du et al., 2019; Allen-Zhu et al., 2019).

Aiming towards a characterization of loss functions exhibiting good stability, we propose a new condition for loss near local minima. This condition, called *Hessian contractive*, is slightly stronger than a general convex condition, but considerably weaker than strongly convex. Formally, the Hessian contractive condition stipulates that near any local minima, (1) the function is convex; and (2) a data dependent Hessian is positive definite in the gradient direction. Theoretically, we show that such a condition is sufficient to guarantee a constant stability bound for SGD (constant step size) near the local minima, while allowing the Hessian to be low rank. We also provide examples showing Hessian Contractive is a reasonable condition for several loss functions. Empirically, we verify the Hessian Contractive condition near a local minima of the loss while training deep neural networks. We sample points from a neighborhood of current iterates by adding Gaussian noise and verify the HC condition locally by Hessian product approximation. Summarizing our second set of contributions:

**R6**. In Observation 1 we show that the family of widely used (convex) linear model loss functions will satisfy the Hessian Contractive condition. One typical example of such linear model loss is the regression loss function. These observation suggests that Hessian Contractive is a condition satisfied by (potentially many) machine learning loss functions.

**R7**. In Theorem 6 we show that the Hessian Contractive condition will localize SGD iterates in a neighborhood of minima, which implies a constant stability bound for SGD near the local minima.

Table 1: Current landscape of stability bounds. [H] indicates results in (Hardt et al., 2016), and * indicates results in this paper. Bounds without [H] or * are trivial. $\beta$ is the smoothness parameter.

| SGD Step Size | Constant $\alpha_t = a/\beta$ | | $\alpha_t = a/(\beta t)$ | Constant $\alpha_t$ |
|---|---|---|---|---|
| *Loss function* | Strongly Convex | Convex | Non-Convex | Hessian Contractive |
| Upper Bound | $O(1)$ [H] | $O(aT/n)$ [H] | $O\left(T^{\frac{a}{1+a}}/n\right)$ [H] $O\left(T^a/n^{1+a}\right)*$ | Theorem 6 |
| Lower Bound | $\Omega(1)$ | $\Omega(aT/n)*$ | Open, evidence* | $\Omega(1)$ |

## 1.1 RELATED WORKS

**Stability and generalization.** The stability framework suggests that a stable machine learning algorithm results in models with good generalization performance (Kearns & Ron, 1999; Bousquet & Elisseeff, 2002; Elisseeff et al., 2005; Shalev-Shwartz et al., 2010; Devroye & Wagner, 1979a;b; Rogers & Wagner, 1978). It serves as a mechanism for provable learnability when uniform convergence fails (Shalev-Shwartz et al., 2010; Nagarajan & Kolter, 2019). The concept of uniform stability was introduced in order to derive high probability bounds on the generalization error (Bousquet & Elisseeff, 2002). Uniform stability describes the worst case change in the loss of a model trained on an algorithm when a single data point in the dataset is replaced. In (Hardt et al., 2016), a uniform stability analysis for *iterative algorithms* is proposed to analyze SGD, generalizing the one-shot version in (Bousquet & Elisseeff, 2002). Algorithmic uniform stability is widely used in analyzing the generalization performance of SGD (Mou et al., 2018; Feldman & Vondrak, 2019; Chen et al., 2018). The worst case leave-one-out type bounds also closely connect uniform stability with *differential private learning* (Feldman et al., 2018; 2020; Dwork et al., 2006; Wu et al., 2017b), where the uniform stability can lead to provable privacy guarantee. While the upper bounds of algorithmic stability of SGD have been extensively studied, the tightness of those bounds remains open. In addition to uniform stability, an *average stability* of the SGD is studied in Kuzborskij & Lampert (2018) where the authors provide *data-dependent* upper bounds on stability[1]. In this work, we report for the first time lower bounds on the uniform stability of SGD. Our tightness analysis suggests necessity of additional assumptions for analyzing the generalization of SGD on deep learning.

**Geometry of local minima.** The geometry of local minima plays an important role in the generalization performance of deep neural networks (Hochreiter & Schmidhuber, 1995; Wu et al., 2017a). The flat minima, i.e., minima whose Hessians have a large portion of zero-valued eigenvalues, are believed to attain better generalization (Keskar et al., 2016; Li et al., 2018). In (Chaudhari et al., 2019), the authors construct a local entropy-based objective function which converges to a solution with good generalization in a flat region, where "flatness" means that the Hessian matrix has a large portion of nearly-zero eigenvalues. However, these observations have not been supported theoretically. In this paper, we propose the Hessian contractive condition that is slight stronger than flat minima. Such condition suggests that the minima is sharp only in the gradient direction while remains flat in other directions, which unifies the geometrical interpretation of flat minima and uniform stability analysis.

## 2 PRELIMINARIES

In this section we introduce the notion of uniform stability and establish our notations. We first introduce the quantities *Empirical and Population Risk* and *Generalization Gap*. Given an unknown distribution $\mathcal{D}$ on labeled sample space $Z = X \times \{-1, +1\}$, let $S = \{z_1, ..., z_n\}$ denote a set of $n$ samples $z_i = (x_i, y_i)$ drawn i.i.d. from $\mathcal{D}$. Let $w \in \mathbb{R}^d$ be the parameter(s) of a model that tries to predict $y$ given $x$, and let $f$ be a loss function where $f(w; z)$ denotes the loss of the model with parameter $w$ on sample $z$. Let $f(w; S)$ denote the *empirical risk* $f(w; S) = E_{z \sim S}[f(w; z)] =$

---

[1]While it is an interesting open problem to get data-dependent lower bounds by lower bounding the average stability, we construct lower bounds on the worst-case stability. Thus our lower bounds are general and not data-dependent.

$\frac{1}{n} \sum_{i=1}^{n} f(w; z_i)$ with corresponding *population risk* $E_{z \sim \mathcal{D}}[f(w; z)]$. The *generalization error* of the model with parameter $w$ is defined as the difference between the empirical and population risks:

$$|E_{z \sim \mathcal{D}}[f(w; z)] - E_{z \sim S}[f(w; z)]|.$$

Next we introduce the Stochastic Gradient Descent (SGD) method. We follow the setting of (Hardt et al., 2016), and starting with some initialization $w_0 \in \mathbb{R}^d$, consider the following SGD update step:

$$w_{t+1} = w_t - \alpha_t \nabla_w f(w; z_{i_t}),$$

where $i_t$ is drawn from $[n] := \{1, 2, \cdots, n\}$ uniformly and independently in each round. The analysis of SGD requires the following crucial properties of the loss function $f(., z)$ at any fixed point $z$, viewed solely as a function of the parameter $w$:

**Definition 1** ($L$-Lipschitz). *A function $f(w)$ is $L$-Lipschitz if $\forall u, v \in \mathbb{R}^d$: $|f(u) - f(v)| \leq L\|u - v\|$.*

**Definition 2** ($\beta$-smooth). *A function $f(w)$ is $\beta$-smooth if $\forall u, v \in \mathbb{R}^d$: $|\nabla f(u) - \nabla f(v)| \leq \beta\|u - v\|$.*

**Definition 3** ($\gamma$-strongly convex). *A function $f(w)$ is $\gamma$ strongly convex if $\forall u, v \in \mathbb{R}^d$:*

$$f(u) > f(v) + \nabla f(v)^\top [u - v] + \frac{\gamma}{2}\|u - v\|^2.$$

**Algorithmic Stability**   Next we define the key concept of *algorithmic stability*, which was introduced by (Bousquet & Elisseeff, 2002) and adopted by (Hardt et al., 2016). Informally, an algorithm is *stable* if its output only varies slightly when we change a single sample in the input dataset. When this stability is *uniform* over all datasets differing at a single point, this leads to an upper bound on the generalization gap. More formally:

**Definition 4.** *Two sets of samples $S, S'$ are twin datasets if they differ at a single entry, i.e., $S = \{z_1, ...z_i, ..., z_n\}$ and $S' = \{z_1, ..., z_i', ..., z_n\}$.*

Consider a possibly randomized algorithm $A$ that given a sample $S$ of size $n$ outputs a parameter $A(S)$. Define the algorithmic stability parameter of $A$ by:

$$\varepsilon_{\text{stab}}(A, n) := \inf\{\varepsilon : \sup_{z \in Z, S, S'} \mathbb{E}_A |f(A(S); z) - f(A(S'); z)| \leq \varepsilon\}.$$

Here $\mathbb{E}_A$ denote expectation over the random coins of $A$. Also, for such an algorithm, one can define its expected generalization error as:

$$GE(A, n) := \mathbb{E}_{S,A}[E_{z \sim \mathcal{D}}[f(A(S); z)] - E_{z \sim S}[f(A(S); z)]].$$

**Stability and generalization:** It was proved in (Hardt et al., 2016) that $GE(A, n) \leq \varepsilon_{\text{stab}}(A, n)$. Furthermore, the authors observed that an $L$-Lipschitz condition on the loss function $f$ enforces a uniform upper bound: $\sup_z |f(w; z) - f(w'; z)| \leq L\|w - w'\|$. This implies that for Lipschitz loss, the algorithmic stability $\varepsilon_{\text{stab}}(A, n)$ (and hence the generalization error $GE(A, n)$) can be bounded by obtaining bounds on $\|w - w'\|$.

Let $w_T$ and $w_T'$ be the parameters obtained by running SGD starting on twin datasets $S$ and $S'$, respectively. Throughout this paper we will focus on the *divergence quantity* $\delta_T := \mathbb{E}_A\|w_T - w_T'\|$. While (Hardt et al., 2016) reports upper bounds on $\delta_T$ with different types of loss functions, e.g., convex and non-convex loss functions, we investigate the tightness of those bounds.

## 3   TIGHTNESS OF EXISTING BOUNDS

In this section we report our main results. We first consider the convex case with constant step size, where we prove 1) that the existing bounds stating $\epsilon_{\text{stab}} \propto t$ are tight, and 2) for linear models, the analysis can be tightened to show that $\epsilon_{\text{stab}}$ does not increase with $t$. Then we move on to the non-convex case, where a) for decreasing step size we improve the existing upper bound, and give evidence that our new upper bound is tight, and b) for constant step size we give loss functions whose divergence $\delta_t$ increases exponentially with $t$.

### 3.1 CONVEX CASE

In this section we analyze the stability of SGD when loss function is convex and smooth. We begin with a construction which shows that Theorem 3.8 in (Hardt et al., 2016) is tight. Our lower bound analysis will require the following quadratic function:

$$f(w; z) = \frac{1}{2} w^\top A w - y x^\top w \tag{1}$$

where $A$ is a $d \times d$ matrix. In the construction of lower bounds, we carefully choose $A$ and $S$ so that the single data point replaced in twin data set will cause the instability of SGD.

**Theorem 1.** *Let $w_t, w'_t$ be the outputs of SGD on twin datasets $S, S'$ respectively. Let $\Delta_t = w_t - w'_t$ and $\alpha_t$ be the step size of SGD. There exists a function $f$ which is convex, $\beta$-smooth, and $L$-Lipschitz on the domain of $w$, and twin datasets $S, S'$ such that:*

$$\mathbb{E}\|\Delta_T\| \geq \frac{L}{3n} \sum_{t=1}^{T} \alpha_t, \;\; and \;\; \varepsilon_{stab} \geq \frac{L}{3n} \sum_{t=1}^{T} \alpha_t. \tag{2}$$

The convex upper bound in Theorem 3.8 in (Hardt et al., 2016) states that $\|\Delta_T\| \leq \frac{L \sum_{i=1}^{T} \alpha_t}{n}$, which implies that the divergence increases throughout training. The lower bound in Theorem 1 suggests the tightness of the upper bound. However, in practice, such phenomenon is not commonly observed, i.e., for a family of convex but not-strongly-convex loss functions, the generalization performance does not deteriorate as the number of training iterations increases. This motivates us to investigate a weaker condition which still can enforce an $O(1)$ stability, without strong convexity. In the next theorem, we restrict ourselves to a family of linear model loss functions and show that the divergence will not increase unboundedly during training.

We shall need the following definition of a $\xi$-self correlated data set. Essentially, a self-correlated dataset requires an average linear dependence of each $x$. Recall that the i'th sample $z_i = (x_i, y_i)$.

**Definition 5.** *A set $S = \{z_1, ..., z_n\}$ is $\xi$-self correlated if $\forall j \in [n], \frac{1}{n} \sum_{i=1}^{n} (x_j^\top x_i)^2 \geq \xi > 0$.*

Assuming that $\forall j \in [n], \|x_j\| \geq r$ for some $r > 0$, definition 5 implies that $S$ is at least $\frac{r^2}{n}$-self correlated. Thus the above condition holds for all datasets $S$ not containing the zero-feature vector. In our next theorem, we leverage on the $\xi$-self correlated condition to prove a non-accumulate uniform stability bound for SGD on a loss function of *Linear Models*. We characterize a linear model by rewriting the loss function $f(w; z)$ in terms of $f_y(w^\top x)$ where $f_y(\cdot)$ is a scalar function depending only on the inner product of the model parameter $w$ and the input feature $x$.

**Theorem 2.** *Suppose a loss function $f(w, z)$ is of the form $f(w, S) = \frac{1}{n} \sum_{j=1}^{n} f_{y_j}(w^\top x_j)$, where $f_y(w^\top x)$ satisfies (1) $|f'_y(\cdot)| \leq L$ , (2) $0 < \gamma \leq f''_y(\cdot) \leq \beta$, (3) $\|x\| \leq R$ and (4) $S, S'$ are $\xi$-self correlated, twin datasets. Let $w_t$ and $w'_t$ be the outputs of SGD on $S$ and $S'$ after $t$ steps, respectively. Let the divergence $\Delta_t := w_t - w'_t$ and $\alpha \leq \frac{1}{\beta}$ be the step size of SGD. Then, $\mathbb{E}\|\Delta_T\| \leq \frac{4LR}{\xi \gamma n}$.*

**Remark 1.** *In (Hardt et al., 2016), an $O\left(\frac{L^2}{\gamma n}\right)$ stability bound is derived on a loss function $f(w, z_i)$ which is strongly convex, i.e., $\nabla^2 f(w) \succ \gamma I$. In practice one can incorporate a strongly convex regularizer to impose strong convexity, often resulting in improved generalization performance in practice (Shalev-Shwartz et al., 2010; Bousquet & Elisseeff, 2002). The $\xi$-self correlated condition allows SGD to maintain a uniformly upper-bounded divergence guarantee for a family of widely used models for arbitrary long training without using strongly convex regularizer. The theorem suggests that if the dataset $S$ is reasonably simple, e.g., every $x_i$ lies in a low dimensional subspace, the divergence of SGD is comparable with a strongly convex loss function. This analysis suggests an alternative condition other can strong convexity can empower SGD an $O(1)$ stability which is data-dependent. This motivates us to go beyond linear model and seek for a generalized condition in Theorem 2. In section 4.1, we propose the Hessian Contractive condition for more general loss function driven by the observation on the linear model.*

**Example: Linear and logistic regression.** Linear regression minimizes the quadratic loss on $w$: $f(w, S) = \frac{1}{2n} \sum_{x_j \in S} (x_j^\top w - y_j)^2$. Note that the Hessian of an individual linear regression loss term is $x_j x_j^\top$ which is *not strongly convex* since it has rank 1. One cannot apply the strongly convex bound,

and the bound for convex suggests stability will increase linearly. However, one can rewrite the loss function as $f_y(w^\top x)$ where $f''_y(\cdot) = 1$. Hence Thm. 2 can be applied to give a non-accumulative bound on SGD's stability. A similar result can be derived for the *logistic regression* loss.

## 3.2 Non-Convex Case

In this section, we construct a non-convex loss function to analyze the tightness of the divergence bound in (Hardt et al., 2016). We first focus on the case of *decreasing step size*.

**Theorem 3.** *There exists a function $f$ which is non-convex and $\beta$-smooth, twin datasets $S, S'$ and constants $a$ s.t. the following holds: if SGD is run using step size $\alpha_t = \frac{a}{0.99\beta t}$ for $1 \le t < T$, and $w_t, w'_t$ are the outputs of SGD on $S$ and $S'$, respectively, and $\Delta_t := w_t - w'_t$, the divergence of SGD after $T$ rounds ($T > n$) satisfies: $\mathbb{E}\|\Delta_T\| \ge \frac{T^a}{3n^{1+a}}$.*

**Comparison to the bound in Theorem 3.12(Hardt et al., 2016)** In (Hardt et al., 2016), an assumption is made on the non-convex loss function, namely that $f(w, z) < 1$. We remark that our function $f$ used in proving the lower bound above does not obey this assumption. Thus for very large $T$, our lower bound may exceed the upper bound in (Hardt et al., 2016), and in general is incomparable due to the lack of this assumption.

Next one observes that in the range $T^{\frac{a}{1+a}} \le n$, the upper bound in (Hardt et al., 2016), namely $O(T^a/n^{1+a})$ is larger than 1, weakening its importance, especially because of the assumption $f(z, w) < 1$, and the fact that when $a$ is small, and one is interested in training faster, smaller values of $T$ in the above range are important. Our divergence lower bound motivates an investigation into the possible tightness of the analysis leading to the upper bound in (Hardt et al., 2016). In the following theorem we prove a *tighter* upper bound for this range of $T$: it does not assume $f(z, w) < 1$, and is non-trivial in the range when $T^{\frac{a}{1+a}} \le n$.

**Theorem 4.** *Assume $f$ is $\beta$-smooth and $L$-lipschitz. Running $T$ ($T > n$) iterations of SGD on $f(w; S)$ with step size $\alpha_t = \frac{a}{\beta n}$, the stability of SGD satisfies: $\varepsilon_{stab} \le \frac{2L^2 T^a}{n^{1+a}}$.*

Dividing our bound with the one in Theorem 3.12 in Hardt et al. (2016), we obtain the ratio $\Omega\left(\frac{T^{\frac{a^2}{1+a}}}{n^a}\right)$. This factor is less than 1 (and so we improve the upper bound) exactly when $T^{\frac{a}{1+a}} \le n$.

Note that this is potentially a large range as $a$ is a small and positive constant. We remark that our tight bound is for *permutation SGD*. The bound for SGD (see Appendix Theorem 4b) using sampling without replacement has an additional $\log(n)$ factor (which we conjecture can be removed), which nevertheless is also a polynomial improvement over the known bounds.

## 4 Deep Learning and Limitations of Bounds

While Theorem 4 improves existing upper bounds on stability by an polynomial factor, it still can not explain the generalization performance of deep learning model. In particular, the analysis relies on a decreasing step size while in training deep learning model, constant instead of a decreasing step size is a common choice. In the next result, we adopt the same construction as in Theorem 3 but with a constant learning rate to show that, unfortunately, SGD may have an exponential divergence rate for general $\beta$ smooth function.

**Theorem 5.** *Let $w_t, w'_t$ be the outputs of SGD on twin datasets $S, S'$, and let $\Delta_t := w_t - w'_t$. There exists a function $f$ which is non-convex and $\beta$-smooth, twin sets $S, S'$ and constants $a, \gamma$ such that the divergence of SGD after $T$ rounds ($T > n$) using constant step size $\alpha = \frac{a}{0.99\beta}$ satisfies: $\mathbb{E}\|\Delta_T\| \ge \frac{1}{n^2} e^{aT/2}$.*

The above theorem implies that the generalization gap may deteriorate at an exponential rate if SGD is run with a constant step size for a non-convex function. However, in Fig. 1a we observe that the distance between the weights of two deep models trained on twin datasets (in other words, the divergence $\delta_T$ *stabilizes after* $\approx 20$ *epochs of training*. Fig. 1a suggests that the generalization gap for deep models may remain constant after sufficiently many training iterations. However, the analysis of (Hardt et al., 2016) suggests that local minima of the loss landscape must be in strongly convex

regions in order to achieve this sort of stability. Indeed, strongly convex Hessians have been proven to be contractive, leading to the stability of SGD (Ge et al., 2015). On the other hand, it is known that the Hessians of the loss functions of common deep learning architectures have a large fraction of eigenvalues which are close to zero (Yao et al., 2019; Chaudhari et al., 2017). This phenomenon requires further investigation to close the gap between theory and practice. Next, we formalize a condition requiring that local minima of deep learning loss landscapes lie in contractive regions, which aids in explaining why the generalization gap stabilizes. We then provide empirical evidence that this property does indeed hold.

## 4.1 GEOMETRY OF LOCAL MINIMA

In Theorems 1 and 5, we show that in general, divergence (and hence stability) may deteriorate at a linear or exponential rate with the number of training iterations when the local minima are not in strongly convex regions. We introduce the following property to aid in understanding the stability of the training curve of SGD.

**Definition 6.** *[Hessian Contractive] For a given set $S = \{z_1, ... z_n\}$, a local minimum $w^*$ is in a $(\sigma, \gamma)$-Hessian contractive region if $\forall w, w'$ with $\max(\|w' - w^*\|, \|w - w^*\|) \leq \sigma$ and $z \in S$,*

$$\nabla_w f(w, z)^\top \mathcal{H}_{w'} \nabla_w f(w, z) \geq \gamma \|\nabla_w f(w, z)\|^2 \tag{3}$$

*where $\mathcal{H}_{w'} = \nabla_w^2 f(w'; S) = \frac{1}{n} \sum_{j=1}^n \nabla_w^2 f(w', z_j)$. When $(\sigma, \gamma)$-Hessian contractive holds for all $\sigma > 0$ and local minima $w^*$, we say $f(w; S)$ is globally $\gamma$-Hessian contractive.*

Hessian contractivity describes a region where the stochastic gradient will lie in the range of a positive semi-definite Hessian matrix. Such a condition prevents the iterates of SGD from escaping from a local minima in the sense that all gradient descent steps will be pulled back to the minima by the power of Hessian, which mimics the structure of a strongly convex loss function but with a weaker assumption. To show that this class is nonempty, we observe the (not strongly convex) linear models loss functions in Theorem 2 satisfy Hessian contractive condition globally.

**Observation 1.** *Suppose loss function $f(w; S)$ conditions in Theorem 2, i.e., $f(w, S) = \frac{1}{n} \sum_{j=1}^n f_{y_j}(w^\top x_j)$ satisfies (1) $|f_y'(\cdot)| \leq L$, (2) $\gamma \leq f_y''(\cdot) \leq \beta$, (3) $\|x\| \leq R$ and (4) $S$ is $\xi$-self correlated, we have $f(w; S)$ is globally $\frac{\xi \gamma}{R^2}$ Hessian Contractive.*

**Dependence on dataset** The Hessian Contractive condition states that the gradient of loss function evaluated on $z$ can be regulated by Hessian evaluated on the whole dataset, which is a generalization of $\xi$-self correlated set condition. In the linear model case, the power of Hessian Contractive condition depends on the the average correlation between data points which indicates the effect of data complexity in generalization. We postulate that the value of $\gamma$ in Hessian Contractive condition can be one complexity measurement of the data, which is believed to affect generalization power (Kawaguchi et al., 2017).

Next, we leverage on the Hessian contractive condition to show that once SGD hits a stationary point around $w^*$ in a $(\sigma, \gamma)$-Hessian contractive region, it will be localized near the minima. The localization of SGD implies that the divergence curve will be stable eventually.

**Theorem 6.** *Given a set $S$ and a $\beta$-smooth and $L$-Lipschitz loss funtion $f$, suppose SGD with a fixed step size $\alpha \leq \frac{\sigma^2}{2\gamma L^2}$ reaches point $w_0$ which is close to a local minimum $w^*$ in a $(\sigma, \gamma)$-Hessian contractive region: $\|w_0 - w^*\| \leq \sigma$ with large enough radius so that $\sigma > \frac{12L}{\gamma}$, then we have $\forall T \geq 1$, $\|w_T - w^*\| \leq \sigma$.*

Theorem 6 states that once the iterate $w$ of SGD encounters a point in a $(\sigma, \gamma)$-Hessian contractive region with large enough radius $\sigma$, $w$ will remain in this region. In terms of the stability of SGD, assuming SGD will bring $w_t$ into one of the local minima quickly (Ge et al., 2015; Allen-Zhu et al., 2019), the divergence $\|w_t - w_t'\|$ will be uniformly bounded by the difference between two local minima $w^*, w^{*'}$ of loss functions $f(w; S)$ and $f(w'; S')$ plus the $\sigma$-radius of the Hessian contractive region. The above suggests that a weaker condition than strong convexity is sufficient to explain the observation in Fig. 1a.

## 4.2 EMPIRICAL SUPPORT FOR HESSIAN CONTRACTIVITY

In order to explain the stabilization of the generalization gap observed when training deep learning models via SGD, we hypothesize that local minima of deep loss landscapes found by SGD lie in Hes-

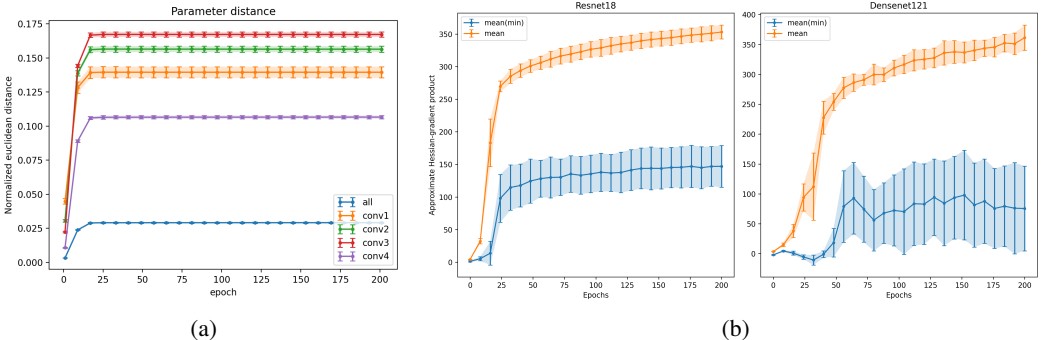

(a)                                                (b)

Figure 1: (a) As in (Hardt et al., 2016), we measure the normalized Euclidean distance between the parameters of two identical deep models (ResNet18) trained on twin datasets (CIFAR10 with a random image removed). We plot the mean over 35 twin datasets (variance shown). The distance first increases but stabilizes after $\approx 20$ train epochs. The "all" curve is the distance between the full model parameter vectors; conv$i$ is the distance between first convolutional layer weights in the $i$-th "block" of ResNet18. (b) Approximate normalized Hessian contractivity. For both neural networks, the "mean" contractivity value at each epoch is averaged over 100 perturbations and 5 different copies of the network trained on CIFAR10. Similarly for "mean(min)", except the min of the 100 approximations after perturbing is taken. Both quantities for both networks provide evidence that SGD converges to an increasingly contractive solution.

sian contractive regions. We propose to empirically support this hypothesis by locally approximating the LHS of Eq. (6)–normalized by $\|\nabla_w f(\tilde{w}, z)\|^2$–during the training of a model, and showing that this quantity steadily increases as training progresses. This indicates that SGD leads the training into regions of increasing contractivity.

We now explain our approximation: For a model $w^*$, we generate many random samples $\tilde{w}$ from the unit sphere around $w^*$. For each $\tilde{w}$, we locally estimate a stochastic version of Hessian-gradient product to get the approximation

$$\nabla_w f(\tilde{w}, z)^\top \mathcal{H}(z) \nabla_w f(\tilde{w}, z) \approx \frac{1}{\eta}[\nabla_w f(\tilde{w}, z) - \nabla_w f(\tilde{w} - \eta \nabla_w f(\tilde{w}, z), z)]^\top \nabla_w f(\tilde{w}, z) \quad (4)$$

where the gradients on the RHS approximation are taken with respect to a random minibatch of samples. For each perturbation $\tilde{w}$ we normalize the corresponding approximation by $\|\nabla_w f(\tilde{w}, z)\|^2$, and then take the mean and minimum over all of these quantities.

For our experiments, we used the well-known CIFAR10 dataset (Krizhevsky et al., 2009) and two popular deep architectures: ResNet18, and DenseNet121 (Huang et al., 2017; Lin & Jegelka, 2018). To match the settings in the theorems of this paper as much as possible and avoid any confounding factors, we avoided standard regularization (e.g. weight decay), data augmentation, and adaptive learning rates. We used SGD with a constant learning rate of 0.01 and momentum of 0.9. The models were trained for 200 epochs and batch size of 128. We trained five copies of each network and checkpointed the models every 8 epochs. For each checkpoint, we generated 100 random perturbations from the unit sphere around the checkpoint weights. We used $\eta = 0.001$ and batch size 128 for the approximation in Eq. (4). We averaged over these 100 approximations and then further took the average over the five networks. The results of these experiments are shown in Fig. 1b, and indicate a clear increase in Hessian contractivity as training progresses.

## 5 CONCLUSION

In this paper, we studied the stability bound of SGD with regard to different types of loss functions. We proved a better upper bound and proved the tightness of various bounds. These tightness results suggest that existing stability bounds may not suffice in answering the generalization problem of deep neural nets. We propose a new Hessian contractive condition that is slighly stronger than convex, but with $O(1)$ stability bound. We provide empirical evidence to support the hypothesis that deep learning loss function is Hessian contractive near local minima.

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

## A    PROOFS

**Lemma 1** (Dynamics of divergence). *Let $f(w; x) = \frac{1}{2}w^\top Aw - yx$. Suppose $[x_i - x_i']/\|x_i - x_i'\|$ is an eigenvector of $A$ where $A[x_i - x_i'] = \lambda_{xx'}[x_i - x_i']$. Let $\Delta_t$ be $w_t - w_t'$, $\alpha_t \leq \lambda_{xx'}$ be the step size of SGD and $\Delta_0 = 0$. Suppose one runs SGD on $f(w, S)$ and $f(w, S')$ where $S, S'$ are twin datasets and $x_i'^\top x_j = 0, x_i^\top x_j = 0, \ \forall j \neq i$, the dynamics of $\Delta_t$ are given by:*

$$\mathbb{E}\|\Delta_{t+1}\| = (1 - \alpha_t \lambda_{xx'})\mathbb{E}\|\Delta_t\| + \frac{\alpha_t}{n}\|x_i - x_i'\| \tag{5}$$

**Proof**: In case the different entry $z_i, z_i$ is not picked, the gradient difference of $f(w; z)$ and $f(w; z')$ is

$$\nabla f(w, z) - \nabla f(w', z') = A[w - w']$$

and in case different entry $z_i, z_i$ is picked,

$$\nabla f(w, z) - \nabla f(w', z') = A[w - w'] + [x_i - x_i']$$

Since $\Delta_0 = 0$, one can inductively show $\Delta_t = \theta_t[x_i - x_i']$ where $\theta_t > 0$. Since SGD selects $z_t = z_t'$ with probability $1 - \frac{1}{n}$ and a different entry with probability $\frac{1}{n}$ we have the following dynamic:

$$\Delta_{t+1} = \begin{cases} (I - \alpha_t A)[w_t - w_t'] & \text{with prob. } 1 - \frac{1}{n} \\ (I - \alpha_t A)[w_t - w_t'] + \alpha_t[x_i - x_i'] & \text{with prob } 1/n. \end{cases}$$

$$\begin{aligned} \mathbb{E}\|\Delta_{t+1}\| =& \mathbb{E}\left[\|\Delta_{t+1}\| | \text{Index } i \text{ is not selected}\right] \mathbb{P}[\text{Index } i \text{ is not selected}] \\ &+ \mathbb{E}\left[\|\Delta_{t+1}\| | \text{Index } i \text{ is selected}\right] \mathbb{P}[\text{Index } i \text{ is selected}] \\ =&(1 - \frac{1}{n})\|(I - A)[w_t - w_t']\| + \frac{1}{n}\|(I - A)[w_t - w_t'] + \alpha_t[x_i - x_i']\| \\ =&(1 - \frac{1}{n})(1 - \alpha_t \lambda_{xx'})\theta_t\|x_i - x_i'\| + \frac{1}{n}[1 - \alpha_t \lambda_{xx'}\theta_t + \alpha_t]\|x_i - x_i'\| \\ =&(1 - \alpha_t \lambda_{xx'})\mathbb{E}\|\Delta_t\| + \frac{\alpha_t}{n}\|x_i - x_i'\| \end{aligned}$$

$\square$

**Lemma 2** (Lower bound on divergence). *Let $\Delta_t$ be $w_t - w'_t$, $\alpha_t$ be the step size of SGD and $\Delta_0 = 0$. Suppose $[x_i - x'_i]/\|x_i - x'_i\|$ is an eigenvector of $A$ where $A[x_i - x'_i] = \lambda_{xx'}[x_i - x'_i]$. Running SGD on $f(w, S)$, we have:*

$$\mathbb{E}\|\Delta_T\| \geq \frac{\|x_i - x'_i\|}{n} \sum_{t=1}^{T-1} \prod_{\tau=t+1}^{T-1} \alpha_t (1 - \alpha_\tau \lambda_{xx'})$$

**Proof**: By Lemma 1 we have

$$\mathbb{E}\|\Delta_T\| = \mathbb{E}\|(I - \alpha_{T-1} A)\Delta_{T-1} + \frac{\alpha_{T-1}}{n}[x_i - x'_i]\|$$

$$= (1 - \alpha_{T-1}\lambda_{xx'})\mathbb{E}\|\Delta_{T-1}\| + \frac{\alpha_{T-1}}{n}\|x_i - x'_i\|$$

$$= \|[x_i - x'_i]\| \frac{1}{n} \sum_{t=1}^{T-1} \alpha_t \prod_{\tau=t+1}^{T-1} (1 - \alpha_\tau \lambda_{xx'})$$

$\square$

**Theorem 1.** *Let $w_t, w'_t$ be the outputs of SGD on twin datasets $S, S'$ respectively, $\Delta_t$ be $w_t - w'_t$ and $\alpha_t$ be the step size of SGD. There exists a function $f$ which is convex and $\beta$-smooth, L-Lipschitz on domain of $w_t, w'_t$ and twin datasets $S, S'$ such that the divergence of the two SGD outputs satisfies:*

$$\mathbb{E}\|\Delta_T\| \geq \frac{L}{2n} \sum_{t=1}^{T} \alpha_t, \ \text{ and } \ \varepsilon_{stab} \geq \frac{L}{2n} \sum_{t=1}^{T} \alpha_t. \tag{6}$$

**Proof** We set $x_i = v, y_i = 0.5$, $x'_i = -v, y'_i = 0.5$. The proof is constructive. Let $f(w, z) = \frac{1}{2} w^\top A w - yx$. The function is $\beta$ smooth and convex by setting $\beta = \|A\|$ and $u^\top A u \geq 0$ for all $u$. We set $S \setminus \{z_i\} = S' \setminus \{z'_i\}$ to lie in the range of $A$, where $A$ is a PSD matrix which is not full-rank. We further set $x_i, x'_i$ to lie in the null space of $A$ so that $Av = 0$. The lower bound $\|\Delta\|_T$ follows from Lemma 2 from the fact that $\Delta_0 = 0$. Since $w^\top A w = w'^\top A w'$, $\varepsilon_{stab} = \sup_z \mathbb{E}|f(w_T, z) - f(w'_T, z)| \geq \mathbb{E}|v^\top[w_T - w'_T]| \geq \frac{1}{n} \sum_{t=1}^{T} \alpha_t$.

The last part we show for any sequence $w_t$ generated from stochastic gradient descent step using data set $S$ will have a constant gradient thus the Lipshitzness will follow. Let $U$ be the range of $A$ (so any $u \in U$ satisfy $u^A u > 0$. Pick $\gamma$ so that $u^\top A u \geq \gamma > 0$ holds for all $u \in U$. For all $u$, $\|w_{t+1}^\top u\| \leq \|(I - \alpha_t A)w_t^\top u + \alpha z^\top u\| \leq (1 - \alpha_t \gamma)\|w_t^\top u\| + \alpha$. This implies $\|w_t^\top u\| \leq \frac{1}{\gamma}$. Now we bound gradient $\|\nabla f_w(w_t, z)\| = \|Aw_t - x\| \leq |v^\top A w| + |v^\top yx| + |u^\top A w| + |u^\top yx| \leq 1 + \frac{\beta}{\gamma}$. This implies that the function is $L$-Lipschitz with $L = 1 + \frac{\beta}{\gamma}$. By setting $\beta = 2\gamma$ the proof follows. $\square$

**Theorem 2.** *Suppose loss function $f(w, z)$ of the form $f(w, S) = \frac{1}{n} \sum_{j=1}^{n} f_{y_j}(w^\top x_j)$ and $f_y(w^\top x)$ satisfies (1) $|f'_y(\cdot)| \leq L$, (2) $0 < \gamma \leq f''_y(\cdot) \leq \beta$, (3) $\|x\| \leq R$ and (4), $S'$ are $\xi$-self correlated. Let $w_t, w'_t$ be the outputs of SGD on twin datasets $S, S'$, $\Delta_t := w_t - w'_t$ and $\alpha \leq \frac{1}{\beta}$ is the step size of SGD. Then,*

$$\mathbb{E}\|\Delta_T\| \leq \frac{4LR}{\xi\gamma n}$$

**Proof**: For simplicity we omit the dependence of $f$ on $y_j$ so that $f_{y_j}(w^\top x_j) = f(w, z_j)$. Note that the gradient of the loss function is $\nabla f_{y_j}(w_t^\top x_j) = f'_{y_j}(w_t^\top x_j)x_j$ and the Hessian is $\nabla^2 f_{y_j}(w_t^\top x_j) = f''_{y_j}(w_t^\top x_j)x_j x_j^\top$. The stochastic gradient step of $f_{y_j}(w_t^\top x_j)$ is

$$w_{t+1} = w_t - \alpha_t f'_{y_j}(w_t^\top x_j)x_j.$$

The dynamics of the divergence can be described as:

$$\mathbb{E}_{1:t+1}\|\Delta_{t+1}\| = \mathbb{E}_{1:t}\|\frac{1}{n} \sum_{j \neq i} \Delta_t - \alpha_t[f'_{y_j}y_j(w_t^\top x_j) - f'_{y_j}(w'^\top_t x_j)]x_j$$

$$\frac{1}{n}(\Delta_t - \alpha_t[f'_{y_i}(w_t^\top x_i)x_i - f'_{y'_i}(w'^\top_t x_i)x'_i])\| \tag{7}$$

Note that $[f'_{y_j} y_j(w_t^\top x_j) - f'_{y_j}(w_t'^\top x_j)]x_j$ can be rewritten as $f''_{y_j}(w_t^{\theta_j \top} x_j)x_j x_j^\top \Delta_t$ where $w^{\theta_j} = (1-\theta_j)w_t + \theta_j w_t', 0 < \theta_j < 1$. Similarly we can also rewrite $f'_{y_i}(w_t^\top x_i)x_i - f'_{y_i'}(w_t'^\top x_i)x_i'$ as

$$
\begin{aligned}
&f'_{y_i}(w_t^\top x_i)x_i - f'_{y_i'}(w_t'^\top x_i)x_i' \\
&= \frac{1}{2}\{f'_{y_i}(w_t^\top x_i)x_i - f'_{y_i}(w_t'^\top x_i)x_i\} \\
&+ \frac{1}{2}\{f'_{y_i'}(w_t^\top x_i')x_i' - f_{y_i'}(w_t'^\top x_i')x_i'\} \\
&+ \frac{1}{2}\{f'_{y_i}(w_t'^\top x_i) + f'_{y_i}(w_t^\top x_i)\}x_i \\
&- \frac{1}{2}\{f'_{y_i'}(w_t^\top x_i') + f_{y_i'}(w_t'^\top x_i')\}x_i' \\
&= \frac{1}{2}f''_{y_i}(w_t^{\theta_i \top} x_i)x_i x_i^\top \Delta_t + \frac{1}{2}f''_{y_i'}(w_t^{\theta'_i \top} x_i')x_i' x_i'^\top \Delta_t \\
&+ \frac{1}{2}\{f'_{y_i}(w_t'^\top x_i) + f'_{y_i}(w_t^\top x_i)\}x_i \\
&- \frac{1}{2}\{f'_{y_i'}(w_t^\top x_i') + f_{y_i'}(w_t'^\top x_i')\}x_i'
\end{aligned}
\tag{8}
$$

Thus equation equation 7 can be written as

$$
\begin{aligned}
\mathbb{E}_{1:t+1}\|\Delta_{t+1}\| = \mathbb{E}_{1:t}\|(I - \frac{\alpha_t}{2n}\sum_{j=1}^{n} f''_{y_j}(w_t^{\theta_i \top} x_i)x_i x_i^\top)\Delta_t \\
+ \frac{\alpha_t}{2n}\{f'_{y_i}(w_t'^\top x_i) + f'_{y_i}(w_t^\top x_i)\}x_i \\
- \frac{\alpha_t}{2n}\{f'_{y_i'}(w_t^\top x_i') + f_{y_i'}(w_t'^\top x_i')\}x_i'\|
\end{aligned}
$$

By the $\xi$-self correlated assumption and letting $\mathcal{H} = \frac{1}{n}\sum_{x_j \in S} x_j x_j^\top$ and $\mathcal{H}' = \frac{1}{n}\sum_{x_j \in S'} x_j x_j^\top$, we have:

$$
\begin{aligned}
\mathbb{E}\|\Delta_{t+1}\| &\leq \mathbb{E}\|(I - \frac{\alpha_t \gamma}{2}\{\mathcal{H} + \mathcal{H}'\})\Delta_t\| + \frac{\alpha_t L}{n}\|x_i\| + \frac{\alpha_t L}{n}\|x_i'\| \\
&\leq (1 - \frac{\alpha_t \xi \gamma}{2})\mathbb{E}\|\Delta_t\| + \frac{2\alpha_t L R}{n}.
\end{aligned}
\tag{9}
$$

The first inequality follows from the fact that $f''(\cdot) \geq \gamma$ and $x_j x_j^\top$s are all PSD. The second inequality follows from the fact that $\Delta_t \in Span\{x_1, .., x_n\}$. Fix $\alpha_t = \alpha$ and the theorem follows. $\qquad\square$

Our next two lemmas are used in the proof of Theorem 3.

**Lemma 3.** *Suppose $x_{t_0} \geq 0$, $x_{t+1} = (1 + \frac{a}{0.99t})x_t + \frac{y}{t}$, we have $x_T \geq y(\frac{T}{t_0})^a$ if $a > 0$ is a sufficiently small constant.*

**Proof**: In the proof we use following inequality:

$$
e^a \leq 1 + \frac{a}{0.99} \leq e^{\frac{a}{0.99}}
$$

where $a > 0$ is a sufficiently small constant.

$$
\begin{aligned}
x_T &= \sum_{t=t_0+1}^{T} \frac{y}{t} \prod_{s=t+1}^{T} (1 + \frac{a}{0.99s}) + x_0 \prod_{t=t_0+1}^{T} (1 + \frac{a}{0.99s}) \\
&\geq \sum_{t=t_0+1}^{T} \frac{y}{t} \exp\left(a \sum_{s=t+1}^{T} \frac{1}{s}\right) \\
&\geq \sum_{t=t_0+1}^{T} \frac{y}{t} \exp\left(a \log(T/t)\right) \\
&\geq y T^a \sum_{t=t_0+1}^{T} \frac{1}{t^{1+a}} \\
&\geq y \left(\frac{T}{t_0}\right)^a
\end{aligned}
$$
(10)

$\square$

**Lemma 4.** *There exists a function $f$ which is non-convex and $\beta$-smooth, twin datasets $S, S'$ and constant $a > 0$ such that the following holds: if SGD is run using step size $\alpha_t = \frac{a}{0.99\beta t}$ for $1 \leq t < T$, and $w_t, w'_t$ are the outputs of SGD on $S$ and $S'$, respectively, and $\Delta_t := w_t - w'_t$, then:*

$$
\forall 1 \leq t_0 \leq T, \quad \mathbb{E}\left[\|\Delta_T\| \big| \Delta_{t_0} \neq 0\right] \geq \frac{1}{2n} \left(\frac{T}{t_0}\right)^a
$$
(11)

**Proof**: Consider the function $f$ in Equation 1, and choose $A$ to have positive and negative eigenvalues. We set the minimum eigenvalue of $A$ equal to $-\beta$ and all other eigenvalues with absolute value at most $\beta$. We select twin datasets for such $A$ as follows. We set all elements in $S \setminus \{x_i\} = S' \setminus \{x'_i\}$ to lie in the column space of $A$. Also, $\forall j \neq i$, choose $x_j$ such that $x_j^\top A x_j > 0$, and choose any $y_j$ between 0 and 1.

Let $v$ be such that $v^\top A v = -\beta$ and $\|v\| = 1$. Finally, let $x_i = v, y_i = 0.5, x'_i = -v, y'_i = 0.5$.

In this setting, one observes that the divergence $\Delta_t$ follows the following dynamic:

$$
\Delta_{t+1} = \left\{
\begin{array}{ll}
(I - \alpha_t A)\Delta_t & \text{with prob. } 1 - \frac{1}{n} \\
(I - \alpha_t A)\Delta_t + \frac{\alpha_t}{2}[x_i - x'_i] & \text{with prob } 1/n.
\end{array}
\right\}.
$$

We first observe that $\Delta_t := w_t - w'_t$ is of the form $v\theta_t$, where $\theta_t > 0$. This can be shown using induction. Let $\tau$ be the first time that $x_i, x'_i$ are picked, we have $\Delta_{\tau+1} = \frac{\alpha_t}{2}[x_i - x'_i] = v\alpha_\tau$. The iterative step of $\Delta_{t+1}$ and $\Delta_t$ implies that $\Delta_{t+1} = v\theta_{t+1}$ where $\theta_{t+1} = (1 + \alpha_t\beta)\theta_t$ with probability $(1 - \frac{1}{n})$ and $\theta_{t+1} = (1 + \alpha_t\beta)\theta_t + \alpha_t$ with probability $\frac{1}{n}$.

The above construction then yields:

$$
\begin{aligned}
\mathbb{E}_{1:t+1}\left[\|\Delta_{t+1}\| \big| \Delta_{t_0} \neq 0\right] &= \mathbb{E}_{1:t}\left[\left(1 - \frac{1}{n}\right)\|(I - \alpha_t A)\Delta_t\| + \frac{1}{n}\|(I - \alpha_t A)\Delta_t + \alpha_t v\|\right] \\
&= \|v\|\mathbb{E}_{1:t}\left[\left(1 - \frac{1}{n}\right)(1 + \alpha_t\beta)\theta_t + \frac{1}{n}((1 + \alpha_t\beta)\theta_t + \alpha_t)\right] \\
&= \|v\|\mathbb{E}_{1:t}\left[[(1 + \alpha_t\beta)\theta_t] + \frac{\alpha_t}{n}\right] \\
&= (1 + \frac{a}{0.99t})\mathbb{E}_{1:t}[\|\Delta_t\| \big| \Delta_{t_0} \neq 0] + \frac{\alpha_t}{n}\|v\|
\end{aligned}
$$
(12)

Now apply Lemma 3, with $x_t = \mathbb{E}[\|\Delta_t\| \big| \Delta_{t_0} \neq 0]$ and $y = \frac{a\|v\|}{0.99\beta n}$. This gives us that $x_T \geq \frac{a\|v\|}{0.99\beta n}(T/t_0)^a = \frac{a}{0.99\beta n}(T/t_0)^a$, since $\|v\| = 1$.

Finally, the claimed bound follows by setting the minimum eigenvalue $\beta = \frac{a}{0.99}$. $\square$

**Theorem 3.** *There exists a function $f$ which is non-convex and $\beta$-smooth, twin datasets $S, S'$ and constants $a$ s.t. the following holds: if SGD is run using step size $\alpha_t = \frac{a}{0.99\beta t}$ for $1 \leq t < T$, and $w_t, w'_t$ are the outputs of SGD on $S$ and $S'$, respectively, and $\Delta_t := w_t - w'_t$, the divergence of SGD after $T$ rounds $(T > n)$ satisfies:*

$$\mathbb{E}\|\Delta_T\| \geq \frac{T^a}{3n^{1+a}} \tag{13}$$

**Proof**: The proof is based on Theorem 4 plus the idea of a "burn-in" period. We have:

$$
\begin{aligned}
\mathbb{E}\|\Delta_T\| &= \mathbb{E}[\|w_t - w'_t\| |\Delta_n = 0]\mathbb{P}[\Delta_n = 0] + \mathbb{E}[\|w_t - w'_t\| |\Delta_n \neq 0]\mathbb{P}[\Delta_n \neq 0] \\
&\geq \mathbb{E}[\|w_t - w'_t\| |\Delta_n \neq 0]\mathbb{P}[\Delta_n \neq 0] \\
&= \left(1 - \left(1 - \frac{1}{n}\right)^n\right) \frac{T^a}{n^{1+a}}\|x_i - x'_i\| \\
&\geq \frac{T^a}{3n^{1+a}}\|x_i - x'_i\|
\end{aligned}
\tag{14}
$$

$\square$

**Lemma 5.** *(Hardt et al., 2016) Assume $f$ is $\beta$-smooth and $L$-lipschitz. Let $w_t, w'_t$ be outputs of SGD on twin datasets $S, S'$ respectively after $t$ iterations and let $\Delta_t := [w_t - w'_t]$ and $\delta_t = \mathbb{E}\|\Delta_t\|$. Running SGD on $f(w; S)$ with step size $\alpha_t = \frac{a}{\beta t}$ satisfies the following conditions:*

- *The SGD update rule is a $(1 + \alpha_t\beta)$-expander and $2\alpha_t L$-bounded.*

- $\mathbb{E}[\|\Delta_t\| |\Delta_{t-1}] \leq (1 + \alpha_t\delta)\|\Delta_{t-1}\| + \frac{2\alpha_t L}{n}.$

- $\mathbb{E}[\|\Delta_T\| |\Delta_{t_k} = 0] \leq \left(\frac{T}{t_{k-1}}\right)^a \frac{2L}{n}.$

**Theorem 4** ( Permutation). *Assume $f$ is $\beta$-smooth and $L$-lipschitz. Running $T$ $(T > n)$ iterations of SGD on $f(w; S)$ with step size $\alpha_t = \frac{a}{\beta t}$, the stability of SGD satisfies:*

$$\mathbb{E}\|\Delta_T\| \leq \frac{2LT^a}{n^{1+a}}, \varepsilon_{stab} \leq \frac{2L^2 T^a}{n^{1+a}} \tag{15}$$

*Proof.* Let $H = t$ represents the event that the first time the SGD pick the different entry is at time $t$:

$$\mathbb{E}\|\Delta_T\| = \mathbb{E}[\|\Delta_T\| |H \leq n]\mathbb{P}[H \leq n] + \underbrace{\mathbb{E}[\|\Delta_T\| |H > n]\mathbb{P}[H > n]}_{0(\text{permutation})}$$

$$
\begin{aligned}
&\leq \frac{1}{n}\sum_{t=1}^{n}\mathbb{E}[\|\Delta_T\| |H = t] \\
&\underset{*}{\leq} \frac{1}{n}\sum_{t=1}^{n}\left(\frac{T}{t}\right)^a \frac{2L}{n} \\
&\leq \frac{2LT^a}{n^2}\int_{t=1}^{n}\frac{1}{t^a}dt \\
&\leq \frac{2LT^a}{n^{1+a}}
\end{aligned}
\tag{16}
$$

The inequality $(*)$ derived by applying Lemma 5. $\square$

**Lemma 6.** *Let $w_t, w'_t$ be outputs of SGD on twin datasets $S, S'$ respectively after $t$ iterations and let $\Delta_t := w_t - w'_t$. Suppose that $t_k = ct_{k-1}$. Then the following conditions hold:*

- $\mathbb{P}[\Delta_{t_{k-1}} = 0|\Delta_{t_k} \neq 0] \leq \frac{n}{n+t_{k-1}}.$

- $\mathbb{P}[\Delta_{t_{k-1}} \neq 0|\Delta_{t_k} \neq 0] \leq \frac{1}{c}\left(1 + \frac{t_k}{n}\right).$

- $\mathbb{E}[\|\Delta_T\| |\Delta_{t_k} \neq 0] \leq \mathbb{E}[\|\Delta_T\| |\Delta_{t_{k-1}} \neq 0]\frac{1}{c}\left(1 + \frac{t_k}{n}\right) + \left(\frac{T}{t_{k-1}}\right)^a \frac{2L}{n}.$

*Proof.* In the proof we will use the following inequality with $r \geq 1$:

$$\frac{n-r}{n} \leq (1 - \frac{1}{n})^r \leq \frac{n}{n+r}$$

i):

$$\begin{aligned}
\mathbb{P}[\Delta_{t_{k-1}} = 0 | \Delta_{t_k} \neq 0] &= \frac{\mathbb{P}[\Delta_{t_{k-1}} = 0, \Delta_{t_k} \neq 0]}{\mathbb{P}[\Delta_{t_k} \neq 0]} \\
&= (1 - 1/n)^{t_{k-1}} \frac{1 - (1 - 1/n)^{t_k - t_{k-1}}}{1 - (1 - 1/n)^{t_k}} \\
&\leq (1 - 1/n)^{t_{k-1}} \leq \frac{n}{n + t_{k-1}}
\end{aligned} \tag{17}$$

ii):

$$\begin{aligned}
\mathbb{P}[\Delta_{t_{k-1}} \neq 0 | \Delta_{t_k} \neq 0] &= \frac{\mathbb{P}[\Delta_{t_k} \neq 0, \Delta_{t_{k-1}} \neq 0]}{\mathbb{P}[\Delta_{t_k} \neq 0]} \\
&= \frac{\mathbb{P}[\Delta_{t_{k-1}} \neq 0]}{\mathbb{P}[\Delta_{t_k} \neq 0]} \\
&= \frac{1 - (1 - 1/n)^{t_{k-1}}}{1 - (1 - 1/n)^{t_k}} \\
&\leq \frac{1 - \frac{n}{n + t_{k-1}}}{1 - \frac{n - t_k}{n}} \\
&\leq \frac{t_{k-1}}{t_k}(1 + \frac{t_k}{n}) \\
&= \frac{1}{c}(1 + \frac{t_k}{n})
\end{aligned} \tag{18}$$

iii):

By applying i) and ii) in the decomposition of $\mathbb{E}[\Delta_T | \Delta_{t_k} \neq 0]$ we have

$$\begin{aligned}
\mathbb{E}[\|\Delta_T\| | \Delta_{t_k} \neq 0] &\leq \mathbb{E}[\|\Delta_T\| | \Delta_{t_{k-1}} \neq 0] \mathbb{P}[\Delta_{t_{k-1}} \neq 0 | \Delta_{t_k} \neq 0] \\
&\quad + \mathbb{E}[\|\Delta_T\| | \Delta_{t_{k-1}} = 0] \mathbb{P}[\Delta_{t_{k-1}} = 0 | \Delta_{t_k} \neq 0] \\
&\leq \mathbb{E}[\|\Delta_T\| | \Delta_{t_{k-1}} \neq 0] \frac{t_{k-1}}{t_k}(1 + \frac{t_k}{n}) \\
&\quad + (\frac{T}{t_{k-1}})^a \frac{2L}{n + t_{k-1}} \\
&= \frac{1}{c}(1 + \frac{t_k}{n}) \mathbb{E}[\|\Delta_T\| | \Delta_{t_{k-1}} \neq 0] \\
&\quad + (\frac{T}{t_{k-1}})^a \frac{2L}{n + t_{k-1}}
\end{aligned} \tag{19}$$

where the last inequality uses the fact that $\mathbb{E}[\|\Delta_T\| | \Delta_{t_k} = 0] \leq (\frac{T}{t_{k-1}})^a \frac{2L}{n}$.  $\square$

**Theorem4b** (Uniformly Independent) Assume $f$ is $\beta$-smooth and $L$-lipschitz. Running $T$ ($T > n$) iterations of SGD on $f(w; S)$ with step size $\alpha_t = \frac{a}{\beta t}$, the stability of SGD satisfies:

$$\mathbb{E}\|\Delta_T\| \leq 16 \log(n) L \frac{T^a}{n^{1+a}}; \quad \varepsilon_{stab} \leq 16 \log(n) L^2 \frac{T^a}{n^{1+a}} \tag{20}$$

*Proof.* We first decompose $\Delta_T$ as follows by selecting $t_k = n$:

$$\mathbb{E}\|\Delta_T\| = \underbrace{\mathbb{E}[\|\Delta_T\| | \Delta_{t_k} = 0] \mathbb{P}[\Delta_{t_k} = 0]}_{\text{Term 1} \leq \frac{2LT^a}{n^{1+a}} \text{ (Lemma 5)}} + \underbrace{\mathbb{E}[\|\Delta_T\| | \Delta_{t_k} \neq 0] \mathbb{P}[\Delta_{t_k} \neq 0]}_{\text{Term 2} \leq \frac{11L \log(n) T^a}{n^{1+a}}} \tag{21}$$

Term 1 is easily bounded by applying Lemma 5 with $\alpha_t = \frac{a}{t\beta}$. To bound Term 2, plug in $\mathbb{P}[\Delta_{t_k} \neq 0] = 1 - (1 - 1/n)^{t_k} \leq \frac{t_k}{n}$ and recursively apply point (iii) from Lemma 6 by setting $t_{i+1} = ct_i$. We get:

$$
\begin{aligned}
& \mathbb{E}[\|\Delta_T\| | \Delta_{t_k} \neq 0] \mathbb{P}[\Delta_{t_k} \neq 0] \\
& \leq \frac{2L}{n} \frac{t_k}{n} \sum_{i=1}^{k-1} (\frac{T}{t_i})^a \frac{n}{n+t_i} \prod_{\tau=i+1}^{k-1} (1 + \frac{t_{\tau+1}}{n}) \frac{t_\tau}{t_{\tau+1}} \\
& \leq \frac{2L}{n} \sum_{i=1}^{k-1} (\frac{T}{t_i})^a \frac{t_{i+1}}{n+t_i} exp(\sum_{\tau=i+1}^{k-1} \frac{t_{\tau+1}}{n}) \\
& \leq \frac{2cL}{n} exp\left(\frac{c}{c-1}\right) \sum_{i=1}^{k-1} (\frac{T}{t_i})^a \frac{t_i}{n+t_i} \\
& \leq \frac{2cLT^a}{n} exp\left(\frac{c}{c-1}\right) \sum_{i=1}^{k-1} \frac{t_i^{1-a}}{n} \\
& \leq \frac{2L\log(n)T^a}{n^{1+a}} \frac{c^a}{\log c} exp\left(\frac{c}{c-1}\right) \\
& \leq \frac{11\log(n)LT^a}{n^{1+a}}
\end{aligned}
\tag{22}
$$

In the second and third inequality we use the fact that $1 + x \leq \exp(x)$ and $t_{i+1} = ct_i$ to get $\prod_{\tau=i+1}^{k-1}(1 + \frac{t_{\tau+1}}{n}) \leq exp(\sum_{\tau=i+1}^{k-1} \frac{t_{\tau+1}}{n}) \leq exp\left(\frac{c}{c-1}\right)$. The last inequality is derived by picking $c = 4$. $\qquad \square$

**Theorem 5.** *Let $w_t, w_t'$ be the outputs of SGD on twin datasets $S, S'$, and let $\Delta_t := w_t - w_t'$. There exists a function $f$ which is non-convex and $\beta$-smooth, twin sets $S, S'$ and constants $a, \gamma$ such that the divergence of SGD after $T$ rounds ($T > n$) using constant step size $\alpha = \frac{a}{0.99\gamma}$ satisfies:*

$$
\mathbb{E}\|\Delta_T\| \geq \frac{1}{n^2} e^{aT/2}
\tag{23}
$$

**Proof**: The proof is similar to Theorem 3. Since $\Delta_t \in Span\{x_i - x_i'\}$, we have:

$$
\mathbb{E}\|\Delta_{t+1}\| \geq (1 - \frac{1}{n})(1 + \alpha_t \beta)\mathbb{E}\|\Delta_t\| + \frac{\alpha_t}{n}\|x_i - x_i\|
$$

Suppose $t_0$ is the hitting time when $\|\Delta_{t_0}\| > 0$ and $\|\Delta_{t_0-1}\| = 0$, $\|\Delta_T\| \geq \frac{\|x_i - x_i'\|}{3n} e^{a(T-t_0)/2}$.

$$
\begin{aligned}
\mathbb{E}\|\Delta_T\| &= \mathbb{E}[\|w_t - w_t'\| | \Delta_1 = 0]\mathbb{P}[\Delta_1 = 0] + \mathbb{E}[\|w_t - w_t'\| | \Delta_1 \neq 0]\mathbb{P}[\Delta_1 \neq 0] \\
&\geq \mathbb{E}[\|w_t - w_t'\| | \Delta_1 \neq 0]\mathbb{P}[\Delta_1 \neq 0] \\
&= \frac{1}{n}(\frac{\|x_i - x_i'\|}{n} e^{aT/2}) \\
&= \frac{\|x_i - x_i'\|}{n^2} e^{aT/2}.
\end{aligned}
\tag{24}
$$

$\qquad \square$

**Theorem 6.** *Given a set $S$ and a $\beta$-smooth and $L$-Lipschitz loss funtion $f$, suppose SGD with a fixed step size $\alpha \leq \frac{\sigma^2}{2\gamma L^2}$ reaches point $w_0$ which is close to a local minimum $w^*$ in a $(\sigma, \gamma)$-Hessian contractive region: $\|w_0 - w^*\| \leq \sigma$ with large enough radius so that $\sigma > \frac{12L}{\gamma}$, then we have $\forall T \geq 1$, $\|w_T - w^*\| \leq \sigma$.*

**Proof**: Let $\mathcal{H}_{w_t, w^*, \theta} = \frac{1}{n} \sum_{z_j \in S} \nabla_w^2 f(\theta w_t + (1 - \theta) w_0, z_j)$ We derive that:

$$
\begin{aligned}
& \|w_{t+1} - w_0\|^2 \\
=& \|w_t - w_0 - \alpha \nabla_w f(w, z)\|^2 \\
=& \|w_t - w_0\|^2 + \alpha^2 \|\nabla_w f(w_s, z)\|^2 - 2\alpha \nabla_w f(w_t, z)^\top [w_t - w_0] \\
=& \|w_t - w_0\|^2 + \alpha^2 \|\nabla_w f(w_s, z)\|^2 - 2\alpha \nabla_w f(w_0, S)^\top [w_t - w_0] \\
& - 2\alpha [\nabla_w f(w_t, z) - \nabla_w f(w_t, S)]^\top [w_t - w_0] \\
& - 2\alpha [\nabla_w f(w_t, S) - \nabla_w f(w_0, S)]^\top [w_t - w_0] \\
\leq& \|w_t - w_0\|^2 + \alpha^2 L^2 - 2\alpha [w_t - w_0]^\top \mathcal{H}_{w_t, w^*, \theta} [w_t - w_0] + 6\alpha L \sigma
\end{aligned}
\tag{25}
$$

Due to the fact that $w_t - w_0 \in Span\{\nabla_w f(w_0, z), ..., \nabla_w f(w_{t-1}, z)\}$, we have

$$
\begin{aligned}
& \|w_t - w_0\|^2 + \alpha^2 L^2 - 2\alpha [w_t - w_0]^\top \mathcal{H}_{w_t, w^*, \theta} [w_t - w_0] + 6\alpha L \sigma \\
\leq& (1 - 2\alpha\gamma) \|w_t - w_0\|^2 + \alpha^2 L^2 + 6\alpha L \sigma
\end{aligned}
\tag{26}
$$

This implies $\|w_t - w_0\| \leq \sigma$ for all $t$. $\qquad\square$

## B    EMPIRICAL SUPPORT FOR THEOREM 6

In order to empirically support Theorem 6, we ran the follow experiment: We started with a fully-trained model $w^*$, and generated random perturbations $\tilde{w} = w^* + \nu$ where $\nu \sim \mathcal{N}(0, \sigma^2)$. We then trained each $\tilde{w}$ until convergence $\tilde{w}^*$, and computed the normalized distance $\|\tilde{w}^* - w^*\|/\|w^*\|$. To match Theorem 6, this distance should be no more than $2\|\tilde{w}^* - w^*\|/\|w^*\|$, i.e., the trained weights $\tilde{w}^*$ after perturbing stay close to $w^*$, as in the conclusion of Theorem 6.

For the above experiment, we first trained both ResNet18 and Densenet121 for 200 epochs, with constant learning rate of 0.01, a momentum of 0.9, and batch size of 128. We generated 20 Gaussian perturbations $\tilde{w}$ of the trained model weights $w^*$ and trained each for 50 more epochs, resulting in weights $\tilde{w}^*$. We used standard deviation $\sigma \in \{0.01, 0.005\}$ for the Gaussian perturbations. We then computed the normalized differences as explained in the previous paragraph. The results are shown in Table 2. Inspecting the table shows that the distances between the trained weights after perturbing are quite close to the distances before further training, with low variance. This indicates that the weights after training the perturbed model are well within the desired range as specified in Theorem 6.

|  |  | $\|\tilde{w} - w^*\|/\|w^*\|$ | $\|\tilde{w}^* - w^*\|/\|w^*\|$ |
|---|---|---|---|
| ResNet | $\sigma = 0.01$ | $0.549 \pm 0.00009$ | $0.557 \pm 0.0003$ |
|  | $\sigma = 0.005$ | $0.274 \pm 0.00005$ | $0.277 \pm 0.007$ |
| DenseNet | $\sigma = 0.01$ | $0.123 \pm 0.00003$ | $0.126 \pm 0.0004$ |
|  | $\sigma = 0.005$ | $0.062 \pm 0.00001$ | $0.065 \pm 0.003$ |

Table 2: Pertubed SGD Results

