# OpenReview forum: "Revisiting the Stability of Stochastic Gradient Descent: A Tightness Analysis"
_ICLR.cc/2021/Conference — Reject_

### Official Review · AnonReviewer1 · 2020-10-24
**Proofs seem not to be rigorous**

**Rating:** 5
**Confidence:** 4

**Review:**

This paper studies stability of SGD which is a popular optimization algorithm. The authors aim to build a tight stability analysis. In particular, the authors show by constructing specific problems that the existing stability bounds for SGD applied to convex problems are tight within a constant factor. Then, the authors refine the stability analysis in non-convex case by presenting new bounds, and show the new bounds are tight. The authors also provide new conditions weaker than strongly convex assumption in both convex and non-convex case. The paper is clearly written and is easy to follow.

Comments:
1. I have doubts in the proof of Lemma 1. It seems that the identity
\[
E\|\triangle_{t+1}\|=(1-\alpha_t\lambda)E[\|\triangle_t\|]+\frac{\alpha_t}{n}\|x_i-x_i'\|
\]
is not correct. The underlying reason is that we can not exchange the expectation and norm. As far as I can see, one can only show
\[
E[\|\triangle_{t+1}\|]=\frac{(n-1)(1-\alpha_t\lambda)}{n}E[\|\triangle_t\|]+\frac{1}{n}E\big\|(1-\alpha_t\lambda)(w_t-w_t')-\alpha_t(x_i-x_i')\big\|
\]
Therefore, I think Lemma 1 may not be right. As a result, the lower bounds on the stability bounds may not be right.


2. The self-corrected condition depends on the dataset S. However, for the uniform stability, one needs to consider all datasets S. Therefore, the divergence bound in Theorem 2 can not be used to get stability bounds. Furthermore, $\xi$ should be zero if one considers all the dataset.

3. In Theorem 4, $\alpha_t=\alpha/(\beta n)$ should be $\alpha_t=\alpha/(\beta t)$? Furthermore, I find it hard to follow the proof. For example, I cannot understand why the authors use the inequality $(1-1/n)^r\leq n/(n-r)$. Is it clear that $(1-1/n)^r\leq 1$? In eq (16), the last third term involves $t_{k-1}$ and $t_k$, while the last second inequality involves only $t_{k-1}$, which is not very meaningful. I also cannot understand well the deduction in eq (20).

4. In the statement of Theorem 6, the authors indicate that $\|w_T-w^\star\|\leq\sigma$. However, the authors only prove this result in expectation. Furthermore, if this inequality only holds in expectation, then one can not use the Definition 6 which requires $w_t$ to be close to $w^*$ almost surely.

In conclusion, although this paper considers a very interesting problem, there are some issues on the correctness of the deduction.

---

> ### Author Response · Authors · 2020-11-17
> **Thank you for your comments**
>
> Answer to Q1:
> Thank you for your comments. We apologize for skipping some steps in the proof. We have added the complete proof to the revised version. Hopefully this increases your confidence in Lemma 1 and our results that follow from it.
>
> Answer to Q2:
> Thanks for pointing this out! As you mention, the minimum of $\xi$ over all datasets could be zero, giving a vacuous bound on the uniform stability in the worst case. However, we feel that the case when the dataset satisfies $\xi$-self correlated condition is significant (happens  when all vectors $x$ in the dataset have norm $ \vert\vert x \vert \vert \geq r$) and therefore worth investigating. If we restrict ourselves to such datasets, then we do obtain an upper bound on the stability. So this result can be seen as a data-dependent bound for a case we think could be interesting. We will make this distinction clear by adding appropriate remarks.
>
> Answer to Q3:
> Thank you very much for pointing out these typos. We have fixed these typos and made the proof more readable.
>
> Answer to Q4:
> Thanks for this observation. We agree that the result is proved only in expectation: the reason is that we did not assume any relation between the variance of the stochastic gradient descent and the radius of the Hessian Contractive region. Assuming a certain correlation between the two, we can indeed prove the result almost surely! We have added this revised claim to the submitted version.

---

### Official Review · AnonReviewer4 · 2020-10-27
**The paper studies the stability of stochastic gradient descent (SGD), which is one of the framework used for explaining generalization.  Overall, I vote for accepting the paper.**

**Rating:** 7
**Confidence:** 3

**Review:**

Summary:

The paper studies the stability of stochastic gradient descent (SGD), which is one of the framework used for explaining generalization. More specifically, the paper investigates the tightness of the algorithmic stability bounds for SGD given by Hardt et al. (2016). Furthermore, the authors propose the Hessian contractive condition, which characterizes deep learning loss functions with good generalization properties, when being trained with SGD.

Pros:

1. The paper concerns the stability-based analysis, which is one of the approaches used for explaining the strong generalization performance of deep neural networks in practice.

2. By investigating the tightness of the stability bound for various types of problem (convex, non-convex), the paper shows that in general, using stability framework seems to hit an obstacle on the way to explaining generalization. Hence, further conditions are needed to guarantee generalization. By this point, the authors propose Hessian contractive condition which is satisfied by potentially many machine learning loss functions and is sufficient to guarantee a better generalization properties.

3. The paper provides empirical evidences / experiments which make the theoretical claims more comprehensible.

Cons:

Apart from the strong points, I still have one concern about the clarity of the paper:

1. The lower bound in Theorem 1 is larger than the upper bound in Theorem 3.8 in Hardt et al. (2016) when the Lipschitz constant is smaller than 1. Can you explain this mismatch?

I hope the authors can address my concern to improve the quality of the paper.

---

> ### Author Response · Authors · 2020-11-17
> **Thank you for your suggestions.**
>
> Answer Q1:
> Thank you for your comments. In our theorem we were implicitly assuming that $L>1$. If $L<1$, we can multiply the lower bound by $L/2$ to make the statement depend on $L$. We have updated the submission to include this case. Thanks for pointing this out.

---

### Official Review · AnonReviewer2 · 2020-10-28
**This paper analyzes the stability of SGD on both convex and non-convex functions based on the seminal work of Hardt et al., 2016.**

**Rating:** 4
**Confidence:** 3

**Review:**

1.	In my understanding, a learning algorithm is stable when a small perturbation in the training does not affect the outcome drastically. Based on this notion, in [Data-Dependent Stability of Stochastic Gradient Descent, Kuzborskij and Lampert, ICML 2018] used (Hardt et al., 2016) and extended them to the distribution-dependent stability setting for both convex and non-convex cases. I also understand that the bounds of (Hardt et al., 2016) are stated in terms of worst-case quantities, but Kuzborskij and Lampert revealed new connections to the data-dependent second-order information. First, among several important works that analyze the stability of SGD, you missed the work of Kuzborskij and Lampert. I suggest that you please include this work, discuss it, and show how you are similar or different than them and how tighter or relaxed your bounds are compared to them. Since both works are based on the seminal work of Hardt et al., 2016, in my opinion, without a proper comparison/discussion this work is incomplete. However, I consider this present manuscript is a fairly well-written manuscript but it is not complete yet.

2.	In the context of uniform stability, [London, B. Generalization bounds for randomized learning with application to stochastic gradient descent, 2016], “partially” addressed how data-independent component such as step-size affects the stability. What is your input on that? Theorem 6 in your manuscript mentions about fixed stepsize but other than that I do not see any discussion on step-size which is in my understanding is an important component in the performance of SGD.

3.	Please correct me if I am wrong: In the proof of Theorem 6 did you use the uniformly bounded gradient? But one can even strongly argue that this bound is actually $\infty$. In my understanding, uniformly bounded gradient with strong convexity leads to a contradiction. That is a stronger argument can be made that the above assumption is in contrast with strong convexity. Please see ["SGD and Hogwild! Convergence Without the Bounded Gradients Assumption" by Nguyen et al.] as one of the instances. How about using more relaxed assumptions such as Strong growth condition on a stochastic gradient as in Assumption 4 of [Dutta et al. AAAI 2020, On the Discrepancy between the Theoretical Analysis and Practical Implementations of Compressed Communication for Distributed Deep Learning]?

---

> ### Author Response · Authors · 2020-11-17
> **Thank you for the comments**
>
> Answer to Q1:
> Thank you for the suggestion. We have added
> this work in the discussion. We also want to stress that the focus of this paper is the  construction of data-independent lower bounds. Constructing a lower bound on the average stability notion of [Data-Dependent Stability of Stochastic Gradient Descent, Kuzborskij and Lampert, ICML 2018] requires more work, and we leave it as an interesting future work.
>
>
> Answer to Q2:
> Perhaps we did not stress this enough, but we have discussed the affects of step size in our work at multiple instances. First, in Table 1 the different columns in row 1 correspond to different step sizes. Also, in Theorem 1, the lower bound of stability is dominated by the summation of step size during the training process. In the non-convex cases, we assume a fixed step size $\alpha/\beta n$ and in theorem 4, we prove that in this setting the upper bound is O($T^{\alpha}/n^{1+\alpha}$).
>
> Answer to Q3:
> We are not assuming convexity or strong-convexity with the Hessian Contractive  condition. In fact, it is quite the contrary: our Hessian Contractive (HC) condition is meant to investigate deep learning loss functions which are non-convex in general. The HC is a local condition, which we postulate aids in generalization. Thus all the contradictions/results that you mention do not apply to our setting, as they assume convexity.

---

### Official Review · AnonReviewer3 · 2020-10-28
**Reviews for Revisiting the Stability of Stochastic Gradient Descent: A Tightness Analysis**

**Rating:** 4
**Confidence:** 4

**Review:**

This paper considers the stability of the stochastic gradient decent algorithm under different conditions. They show a lower bound for the stability of SGD in the smooth and convex case, and show that the bound can be tightened for linear models. They give a tight bound for the stability of SGD with decreasing step size in the non-convex case.  Then they propose the Hessian Contractive condition, and under this condition a tight bound for the stability of SGD with constant step size is given.

pros: 1, They propose the Hessian Contractive condition which is weaker than strongly convex and show that the family of widely used (convex) linear model loss functions will satisfy the Hessian Contractive condition.

2, They analyzed the stability of SGD and give the lower and upper bounds for the stability in many cases.

cons: 1, In Theorem 3, they give a lower bound for the stability of SGD with decreasing step size in the non-convex case. However, this lower bound is larger than the upper bound in Hardt 2016 when T goes to infinity and the other parameters fixed. This contradiction implies that the results in Theorem 3 should be incorrect.

2, They claim the O(1) uniform stability of SGD in under the Hessian Contractive condition. But the uniform bound of $||w^* - w^{*\prime}||$ is not proved.

After the rebuttal.

The authors partially addressed my concerns. I have read other reviewers' comments. I decide to remain the current score.

---

> ### Author Response · Authors · 2020-11-17
> **Thank you for your comments**
>
>
> Answer to Q1.  Thank you for your comment. Our lower bound in the non-convex case is for the \textit{divergence} of the SGD, not the uniform stability. For the divergence of SGD, the upper bound provided in Hardt 2016 is $O\left( \frac{T^{a}}{n} \right)$, which is larger than our lower bound of $O\left( \frac{T^a}{n^{1+a}}  \right)$. Thus there is no contradiction.
>
> Answer to Q2. Agreed; that bound was stated in the table without an accompanying theorem. We will retract that claim for now, and try to add it later.

---

> > ### Comment · AnonReviewer3 · 2020-11-18
> > **Thank you for the reply**
> >
> > 1, Under Theorem 3, it is claimed clearly that the bound in Hardt 2016 is $O(\frac{T^{\frac{a}{1+a}}}{n})$ which does not match the lower bound in Theorem 3. If the divergence bound in Hardt 2016 is $O(\frac{T^a}{n})$, please specify where it comes from and transform the parameters if necessary.
> >
> > 2, In the non-convex case with decreasing step size, when $T$ is large, the upper bound is actually
> > larger than that of Hardt 2016 in Table 1. This may make the result less important.

---

> > > ### Author Response · Authors · 2020-11-25
> > > **We address both questions in the paragraph after Thm 3**
> > >
> > > Thank you for raising this subtle but important point. We address both questions in the paragraph after Thm 3.
> > >
> > > **Answer to Q1:**  In Hardt et al. 2016 , an assumption is made on the non-convex loss function, namely that $f(w,z)<1$. We remark that our function $f$ used in proving the lower bound in Theorem 3 does not obey this assumption. Thus for very large $T$, our lower bound may exceed the upper bound in Hardt et al. 2016 , and in general is incomparable due to the lack of this assumption.
> > >
> > >
> > >
> > >
> > >
> > > **Answer to Q2 :**
> > > We observe that in the range $T^{\frac{a}{1+a}} \leq n$,  the upper bound in Hardt et al. 2016 , namely $O(T^{a}/n^{1+a})$ is larger than $1$, weakening its importance, especially because of the assumption $f(z,w)<1$, and the fact that when $a$ is small, and one is interested in training faster, smaller values of $T$ in the above range are important. Our divergence lower bound motivates an investigation into the possible tightness of the analysis leading to the upper bound in Hardt et al. 2016 . In the Theorem 4 we  prove a tighter upper bound for this range of $T$: it does not assume $f(z,w)<1$, and is non-trivial in the range when $T^{\frac{a}{1+a}} \leq n$.

---

### Decision · Program_Chairs · 2021-01-07
**Final Decision**

**Decision:**

Reject

**Comment:**

The AC and reviewers agree this is an important line of research. However, only one reviewer was initially positive, as the other reviewers raised some issues, and the rebuttal only partially addressed some of them (e.g., the reviewer is now OK with Lemma 1 being correct), but there are typos in the proofs, and there were other issues like the uniform bound that R3 brought up which was retracted in the revision, and such. These issues give us a bit of lack of confidence in the rigor of all the results.

In addition to the lack of carefulness in places, this paper (more than usual for an accepted paper) seemed to miss references in the literature. In addition to all the ones pointed out in the reviews (especially R3's, which I don't think was fully adequately discussed in the rebuttal), other tight lower bounds on uniform stability have been developed recently (see Thm 4.2 in Bassily et al. https://arxiv.org/pdf/2006.06914.pdf).  From the optimization point of view, it is undesirable to introduce new conditions unless really necessary, and often these new conditions are previously known under a different name; if they really are new, they should be compared to old conditions. In particular, then new "Hessian contractive condition" should be compared to standard non-convex conditions like strong growth, error bound, Polyak-Lojasiewicz, etc.

Finally, this is based off the Hardt/Recht/Singer 2016 paper, but there is a more recent Hard/Recht work that argues that algorithmic stability is not the right tool, because it cannot explain the fact that training error drops roughly the same with real data or with data with completely random labels -- so any generalization theory has to be data dependent. See: "Understanding deep learning requires rethinking generalization" (https://arxiv.org/abs/1611.03530) Chiyuan Zhang, Samy Bengio, Moritz Hardt, Benjamin Recht, Oriol Vinyals.  So this issue should be addressed as well.

Overall, this could be a promising paper and the AC recommends the reviewers make a substantial revision addressing these concerns.